# Maternal Fatty Fish Intake Prior to and during Pregnancy and Risks of Adverse Birth Outcomes: Findings from a British Cohort

**DOI:** 10.3390/nu11030643

**Published:** 2019-03-16

**Authors:** Camilla Nykjaer, Charlotte Higgs, Darren C. Greenwood, Nigel A.B. Simpson, Janet E. Cade, Nisreen A. Alwan

**Affiliations:** 1Nutritional Epidemiology Group, School of Food Science and Nutrition, University of Leeds, Leeds LS2 9JT, UK; J.E.Cade@leeds.ac.uk; 2School of Biomedical Sciences, Faculty of Biological Sciences, University of Leeds, Leeds LS2 9JT, UK; 3Department of Obstetrics and Gynaecology, University of Leeds, Leeds LS2 9JT, UK; charlotte.higgs@nhs.net (C.H.); N.A.B.Simpson@leeds.ac.uk (N.A.B.S.); 4Division of Biostatistics, Centre for Epidemiology and Biostatistics, University of Leeds, Leeds LS2 9JT, UK; D.C.Greenwood@leeds.ac.uk; 5School of Primary Care and Population Sciences, Faculty of Medicine, University of Southampton, Southampton General Hospital, Southampton SO16 6YD, UK; N.A.Alwan@soton.ac.uk; 6NIHR Southampton Biomedical Research Centre, University of Southampton and University Hospital Southampton NHS Foundation Trust, Southampton SO16 6YD, UK

**Keywords:** fatty fish, essential fatty acids, omega-3, pregnancy, birth weight, foetal growth, preterm birth

## Abstract

Fish is an important source of the essential fatty acids contributing to foetal growth and development, but the evidence linking maternal fatty fish consumption with birth outcomes is inconsistent. In the UK, pregnant women are recommended to consume no more than two 140 g portions of fatty fish per week. This study aimed to investigate the association between fatty fish consumption before and during pregnancy with preterm birth and size at birth in a prospective birth cohort. Dietary intake data were acquired from a cohort of 1208 pregnant women in Leeds, UK (CARE Study) to assess preconception and trimester-specific fatty fish consumption using questionnaires. Multiple 24-h recalls during pregnancy were used to estimate an average fatty fish portion size. Intake was classified as ≤2, >2 portions/week and no fish categories. Following the exclusion of women taking cod liver oil and/or omega-3 supplements, the associations between fatty fish intake with size at birth and preterm delivery (<37 weeks gestation) were examined in multivariable regression models adjusting for confounders including salivary cotinine as a biomarker of smoking status.. The proportion of women reporting any fatty fish intake decreased throughout pregnancy, with the lowest proportion observed in trimester 3 (43%). Mean intakes amongst consumers were considerably lower than that recommended, with the lowest intake amongst consumers observed in the 1st trimester (106 g/week, 95% CI: 99, 113). This was partly due to small portion sizes when consumed, with the mean portion size of fatty fish being 101 g. After adjusting for confounders, no association was observed between fatty fish intake before or during pregnancy with size at birth and preterm delivery.

## 1. Introduction

Preterm birth (<37 weeks gestation) and low birth weight (LBW) are important determinants of neonatal mortality and morbidity. They are also linked with higher risks of metabolic, neurological and cardiovascular disease in adult life [1,2,3]. Maternal nutrition during pregnancy has been postulated to play a role in the prevention of these adverse birth outcomes [4,5].

Recent research has focused on the role of fatty acids, in particular the omega-6 and omega-3 long chain polyunsaturated fatty acids (LCPUFA), which are derived from their respective precursors, linoleic (LA) and linolenic (LNA) acids. These are vital for the development of cell membranes and new tissues [6,7,8] and are classified as essential fatty acids (EFA) as they can only be derived from the maternal diet. During pregnancy the most biologically active LCPUFAs, docosahexaenoic acid (DHA) and arachidonic acid (AA) have been shown to have beneficial effects [9], particularly on the development of the foetal brain and retina [8]. These EFA cannot be synthesised in the human body [6,7,10] and the conversion rate of precursor to LCPUFA derivative within the foetus is limited [11]. Consequently, the foetus is heavily dependent on the maternal diet for EFA through transport across the placenta [11,12,13]. Additionally, as the EFA status of the mother has been found to decline during pregnancy [6,14], a dietary source is paramount for meeting the demand for maternal-foetal exchange.

Fish are an important source of essential LCPUFAs, particularly the n-3 PUFAs. However, the extent to which fish intake plays a role in shaping pregnancy outcomes is unclear, as evidence regarding maternal consumption and birth outcomes is inconclusive. Findings from some birth cohorts suggest a positive association between total fish intake and birth weight [15,16,17,18,19], with women with a high total fish intake being less likely to have low birthweight (LBW) babies [16,19] as well as preterm birth [20,21]. However, negative associations have also been found [15,18,22,23,24] and in some cases no association with preterm birth [15,17,23,24,25] nor size at birth [15,24,25,26,27] has been evident.

It has been hypothesised that adverse associations may be due to contaminants in fish including mercury and persistent organic pollutants (POPs). Fatty fish is a known source of these contaminants, particularly in larger fish species [22]. However, studies that have focused on differentiating between types of fish consumed including lean, fatty and shellfish in relation to birth outcomes have been inconclusive [15,18,19,24,27] although there may be a trend toward a negative association between fatty fish and foetal growth [18,22].

The current advice in the UK is to consume at least two portions of fish/week (~140 g/portion), one of which should be fatty fish [28]. This recommendation also applies to pregnant women and women trying to conceive but with an upper limit of maximum two portions of fatty fish/week. Pregnant women and women trying to conceive are also advised to avoid consumption of larger species such as marlin, swordfish and shark [28]. Despite the guideline stating that intake of up to two portions of fatty fish/week does not present any harm, many Western pregnant women consume limited amounts of fish [10,29,30] resulting in low intakes of LCPUFA which could be potentially detrimental to foetal development.

Using data from a prospective UK-based birth cohort (the Caffeine and Reproductive Health study (CARE)) [31], this paper aimed to estimate maternal fatty fish intake frequency and portion size before and during pregnancy, and investigate the association between maternal fatty fish intake before and during pregnancy with both preterm birth and size at birth.

## 2. Materials and Methods

### 2.1. Participants and Study Design

The CARE study is a British prospective birth cohort with the primary aim of investigating the associations between caffeine in the maternal diet and pregnancy outcomes [31]. Between 2003 and 2006, low risk pregnant women aged 18–45 years were recruited from Leeds Teaching Maternity Hospitals at 8–12 weeks gestation. A total of 5959 women were considered, of whom 4571 met the inclusion criteria. Eligible women were sent detailed information about the study and 1374 consented to participate. All participants gave written, informed consent and the study was conducted in accordance with the Declaration of Helsinki and approved by the Leeds West Local Research Ethics Committee (reference number 03/054).

### 2.2. Assessment of Maternal Fatty Fish Intake

#### 2.2.1. Recall Data

Rather than using the Scientific Advisory Committee on Nutrition (SACN) estimate of 140 grams (g) per portion of fatty fish [28], which is based on data from a non-pregnant population (the National Diet and Nutrition Survey) we derived an estimate of the average portion size of fatty fish from 24 h dietary recalls administered by research midwives at 14–16 and 28 weeks gestation. To get a better picture of usual consumption throughout pregnancy an average portion size of fatty fish was derived for women who consumed fatty fish at both recalls. Participants were asked to record all food and drink consumed in a 24 h period (12 midnight to 12 midnight), including portion size and drink amounts. An example recall was provided as guidance. Reported canned tuna intake was removed from the analysis as it is not considered a fatty fish due the majority of the fat content being removed during the canning process [28].

#### 2.2.2. Self-Reported Questionnaires

Fatty fish consumption was ascertained prior to and throughout pregnancy using a frequency type self-reported questionnaire adapted from the UK Women’s Cohort Study [32] and administered at enrolment (12–18 weeks gestation), assessing consumption in the 4 weeks leading up to pregnancy and trimester 1; week 28, assessing trimester 2 consumption and postpartum (weeks 46–50) assessing trimester 3 consumption. Participants were asked how often (never; less than once/month; 1–3 times/month; once/week; 2–4 times/week; 5–6 times/week; once/day; 2–3 times/day; 4–5 times/day and >6 times/day) they consumed fatty fish (examples given were: salmon, tuna (fresh only), herring, kipper, mackerel, pilchards, sprats and swordfish). No examples of what constitutes a portion were given in the questionnaire. Frequency of fish consumption derived from the questionnaires was converted to times per week, which was then multiplied by the portion estimate of fish obtained from the recall data (see above) in order to get weekly consumption in grams for each of the trimesters.

### 2.3. Assessment of Pregnancy Outcomes

Information regarding pregnancy outcomes was collected from hospital maternity records. The two primary outcomes assessed were preterm birth (defined as <37 weeks gestation) and small for gestational age (SGA), defined as <10th individualised birth centile taking into account gestational age, maternal height, weight, ethnicity, parity, child sex and birth weight [33]. Actual birth weight was also analysed as a secondary measurement and expressed as a continuous variable in grams and a binary variable LBW defined as <2500 g. Duration of gestation was calculated from the date of the last menstrual period, and confirmed by ultrasound scans dating at around 12 and 20 weeks gestation.

### 2.4. Assessment of Participant Characteristics

Maternal characteristics including age, ethnicity, pre-pregnancy weight, height, parity, education (university degree versus no degree) were self-reported in the preliminary administered questionnaire. Caffeine intake (mg/day) and alcohol consumption (units/day) were assessed throughout pregnancy using the same frequency type questionnaire used to assess fatty fish intake. Smoking status was objectively measured using salivary cotinine levels at enrolment. Participants were classified based on cotinine concentrations as active smokers (>5 ng/mL), passive smokers (1–5 ng/mL) or non-smokers (<1 ng/mL). Daily total energy intake was derived from the 1st 24 h of food recall data.

### 2.5. Statistical Power Calculation

Comparing mothers consuming > 2 portions/week to non-consumers, the study had 80% power to detect an odds ratio of approximately 0.4 for SGA. The equivalent test for linear trend including the intermediate category half way between these extremes would have 90% power. Similarly, comparing the birth weight of babies born to mothers consuming > 2 portions of fish/week with non-consumers, assuming the SD to be approximately 500 g, this study had 85% power to detect a difference of 150 g in birth weight at *p* < 0.05.

### 2.6. Statistical Analysis

Analysis was conducted using the continuous weekly fish variable assigned into three categories of intake based on the current UK guidelines of no more than 2 portions of fatty fish per week [28] with the addition of a “no fish” category which was used as the referent group: no fish, ≤2 portions/week and >2 portions/week.

Univariable analyses were performed using one-way ANOVA for normally distributed variables, Kruskal-Wallis for non-parametric variables and chi-squared test for categorical outcomes. Multivariable linear and logistic regression models were used to assess the association between maternal fatty fish intake and continuous and dichotomous birth outcomes respectively. Maternal pre-pregnancy weight, height, ethnicity, parity, gestation and neonatal sex were accounted for when calculating the SGA variable and were adjusted for in the preterm delivery (omitting gestation) and birth weight models. Covariates adjusted for in all models were selected based on *a priori* knowledge from the literature and included maternal age, salivary cotinine levels, self-reported caffeine intake and alcohol consumption and university degree status as a marker for socioeconomic status.

In order to separate the effect of fatty fish from supplements as opposed to dietary sources on birth outcomes, women taking any cod liver oil and/or omega-3 supplements were removed from the analysis. Women with extreme values for energy intake (highest 1% and lowest 1%), obtained from the 24 h recall data, were excluded due to possible bias with self-reported dietary intake, as proposed by Meltzer et al. [34].

Sensitivity analyses were conducted taking into account previous high-risk pregnancies (including a previous LBW baby, previous gestational diabetes (GDM) and previous gestational hypertension (GHT)) and total energy intake during pregnancy. Sensitivity analyses were also done by excluding women who developed GHT and GDM during their current pregnancy (3rd trimester only). All analyses were performed using the Stata 14 software (Stata, College Station, TX, USA).

## 3. Results

Of the 1374 mothers who consented, 1303 agreed to participate and were enrolled into the CARE study. Of these, nine were lost to follow-up, five terminated pregnancies and others were excluded due to stillbirth (*n* = 6), neonatal death (*n* = 3) and late miscarriage (*n* = 10). Following exclusions of women taking cod liver oil and/or omega-3 supplements (*n* = 37) as well as those with extreme energy intakes (*n* = 25) left 1208 mothers with data available on birth outcomes.

### 3.1. Estimation of Portion Size and Types of Fatty Fish Consumed (24-h Recall)

A total of 1276 women reported dietary information by recall at weeks 14–16, and 601 women at week 28. Of these women, 162/1276 (13%) and 70/601 (12%) reported intakes of any fatty fish at the 1st and 2nd recall respectively. Combining both sets of recall data together, a total of 106 women reported fatty fish intake during the 1st and 2nd recall. The amount of fish consumed in grams at each meal was used to obtain an average portion size of 101 g (min: 10 g, max: 300 g).

Of the 106 women consuming fatty fish in the 24 h recall data (14–28 weeks gestation), 52 (49%) women ate salmon, 25 (24%) ate raw tuna and 14 (13%) ate mackerel. Other types of fatty fish included anchovies (5%), sardines (7%), trout (6%) and orange roughy (0.9%). Fatty fish consumption accounted for 4.8% of the total energy intake.

### 3.2. Frequency of Fatty Fish Consumption (Questionnaire)

Of the 1208 women with birth outcome data, 1116 (92%) women had information available on frequency of fatty fish intake before pregnancy, 1114 (92%) in the 1st trimester, 812 (67.2%) in the 2nd trimester and 409 (34%) in the 3rd trimester (Table 1). For those women who reported consuming any fatty fish, intake before pregnancy (123.5 g/week) was significantly higher (*p* < 0.0001) than trimester 1 & 2 (106.4 and 107.4 g/week respectively) but slightly lower than the mean intake in the 3rd trimester (136.5 g/week). The proportion of women reporting any fatty fish intake, however, decreased throughout pregnancy with the lowest proportion observed in trimester 3 (43%).The prevalence of women consuming within the recommended guidelines of no more than 2 portions of fatty fish per week was highest in trimester 1 (47%) and in the 2nd trimester (49%), with mean intakes for women reaching 64.3 g (95% CI 61.0 to 67.7) and 71.3 g (95% CI 66.6 to 75.7) per week, respectively.

### 3.3. Maternal Characteristics According to Categories of Fish Intake

Table 2 shows characteristics of participants according to maternal fatty fish intake in trimester 1. Women who consumed fatty fish during pregnancy were more likely to be older, have a university degree, to consume alcohol, were less likely to smoke and less likely to live in an area within the most deprived Index of Multiple Deprivation (IMD) quartile. These characteristics were consistent across all trimesters and the four weeks leading up to pregnancy. Women consuming fish in trimester 1 & 2 were also more likely to have a lower BMI, and those consuming fish in trimester 1 were shown to have a lower caffeine intake than non-fish consumers.

### 3.4. Pregnancy Outcomes

Of the 1208 women with information on birth outcomes, 44 babies (4%) were delivered preterm (<37 weeks gestation) with a mean gestational age of 34.29 weeks (SD = 2.99). A further 153 (13%) babies were born SGA (<10th centile) and 46 (4%) were LBW (<2500 g); the latter of which 27 (56%) were born preterm. The mean birth weight of the total sample was 3446 g (SD = 537 g).

### 3.5. Relationship between Fish Intake before Pregnancy and Birth Outcomes

There was no evidence of an association between fatty fish intake before pregnancy and preterm birth nor size at birth (Table 3).

### 3.6. Relationship between Fish Intake during Pregnancy and Preterm Birth

Compared to mothers consuming no fish in trimester 1, in unadjusted analysis mothers consuming up to 2 portions and >2 portions of fatty fish/week were less likely to have babies born preterm (OR: 0.5, 95% CI: 0.3, 1.0 & OR: 0.3, 95% CI: 0.1, 1.1 respectively; *p*_trend_ = 0.05) (Table 4). After adjustment for potential confounders, these estimates were largely unchanged, though no longer statistically significant (OR: 0.6, 95% CI: 0.3, 1.3 & OR: 0.3, 95% CI: 0.1, 1.3 respectively, *p*_trend_ = 0.2). A similar trend could be observed in trimester 2 for mothers consuming ≤2 portions of fatty fish/week compared to non-consumers (OR: 0.4, 95% CI: 0.2, 0.9; *p*_trend_ = 0.06), becoming non-significant after adjustment (OR: 0.5, 95% CI: 0.2, 1.2, *p*_trend_ = 0.2). There was no association between fatty fish intake in the 3rd trimester and preterm birth.

### 3.7. Relationship between Fish Intake during Pregnancy and Size at Birth

When comparing babies born to mothers consuming no fatty fish in trimester 1, mothers consuming up to two portions of fatty fish/week had babies weighing 58.4 g less (95% CI: −115.1, −1.5) although there was no linear trend (*p*_trend_ = 0.1). There was no evidence of any relationship between fish intake in the second or third trimester and size at birth expressed as birth weight (g), SGA (<10th centile) or low birth weight (Table 4).

### 3.8. Sensitivity Analysis

Adding total energy intake to the regression models did not affect the results. Similarly, including an indicator for high risk pregnancies as a possible moderator (*n* = 175) did not significantly alter findings nor did excluding women who developed GDM (*n* = 3) or GHT (*n* = 19).

## 4. Discussion

As far as we are aware this is the only British prospective birth cohort study assessing maternal fatty fish intake prior to and throughout each of the trimesters separately in relation to pregnancy outcomes.

The results showed the majority of pregnant women were consuming considerably less than two portions of fatty fish per week prior to and throughout pregnancy and a trend towards a decreased fatty fish consumption with the progression of pregnancy. Within this study there was no evidence of a statistically significant association between maternal fatty fish intake and gestational length and size at birth, when taking known confounders into account.

### 4.1. Fish Intake and Maternal Characteristics

The proportion of mothers reporting any fatty fish consumption decreased as pregnancy progressed. Among consumers, mean weekly intakes were highest for the period leading up to pregnancy and the 3rd trimester (124 g and 137 g/week respectively) but still considerably lower than the mean of 190 g of fatty fish/week reported in a UK national survey of women (non-pregnant women aged 19–64) carried out around the same time [35] and noticeably lower than the UK guidelines of up to two portions of 140 g fatty fish/week.

The proportion of women in our study not consuming any fatty fish in the 3rd trimester (56.7%) was slightly higher compared to results from the Avon Longitudinal Study of Parents and Children (ALSPAC) which showed in their study of fish intake in pregnancy and birth weight that 42.6% of pregnant women (*n* = 11,511) reported never or rarely consumed any fatty fish in the 3rd trimester [17]. Compared to other non UK studies assessing fatty fish intakes in Western pregnant women, the proportion not consuming any fatty fish were 33% during the 1st trimester in a Dutch birth cohort (*n* = 3380) [25], 11% during the 2nd trimester in a large Norwegian birth cohort (*n* = 62,099) [19] and 24% reported consuming <0.2 portions of fatty fish/month before pregnancy in a US cohort [27], all lower than that observed in our cohort. Results from the Danish National Birth Cohort (DNBC) however (*n* = 44,824) reported a similar proportion of 54% of non-consumers from their assessment of fatty fish intake in the 2nd trimester [22]. Similarly, results from a Spanish cohort of pregnant women (INMA) showed 41% of women reporting consuming <1 portion of fatty fish/month [18]. Results from a meta-analysis by Leventakou et al., (2014) of 19 European cohorts (some of which are mentioned above) showed a considerable variation in fatty fish intake between countries; with Italian, Spanish and Portuguese mothers consuming fatty fish more than twice as often as Irish & French mothers. It is however impossible to tell how much more fatty fish the Spanish mothers ate than the Irish mothers, for instance, because the researchers had data only on frequency, not quantity [36].

Although it is probable that some women simply do not like fish, reasons for low consumption are likely to include perceptions about cost, access to stores that sell fish, and uncertainty about preparation and cooking methods. Furthermore, some women may abstain from eating fish out of a worry that they and their babies will be harmed by contaminants present in some types of fish, a concern which is highlighted in the current UK guidelines but may actually result in a lack of consumption rather than a lowered intake of fatty fish. The characteristics of the mothers in our study across categories of increased fatty fish consumption are consistent with those observed in other studies where slightly older women, those consuming alcohol and women of higher socioeconomic status and higher education tended to consume higher levels of fish and were less likely to be smokers [16,17,19,22,23,25,26].

### 4.2. Interpretation of Main Findings

We did not find any evidence of an association between maternal fatty fish intake before and during pregnancy with gestational age or size at birth.

In another British birth cohort (ALSPAC), Rogers et al., (2004) used n-3 fatty acids as a marker of fish consumption and found no association with preterm birth, LBW or intrauterine growth retardation once they adjusted for confounders [17]. Despite having data on type of fish consumed they did not relate this to birth outcomes but focused instead on n-3 fatty acid intake from fish as well as frequency of total fish consumption making it impossible to make direct comparisons to our study. Other studies have reported a similar lack of association between maternal fatty fish intake and birth outcomes [18,19,24,27]. In their meta-analysis Leventakou et al. (2014) in addition to assessing total fish intake, also assessed types of fish (fatty, lean and seafood) in relation to birth outcomes and similarly to our results, they found no association between fatty fish and gestational age or LBW. Where lean fish and shellfish had no significant associations with any birth outcomes, they did observe a positive association between fatty fish and birth weight, albeit a small one at 2.38 g (95% CI: 0.51, 4.25) for every 1 unit (times/week) increment. The authors stipulated that the n-3 LCPUFA content in fatty fish could be the contributing factor behind the overall positive association they found between total fish intake and birth weight [36]. Contrary to this, Halldorson et al. (2007) reported a reduction of 27.5 g in birthweight of babies born to mothers consuming fatty fish more than four times/month compared to non-consumers as well as an increased risk of having babies born SGA [22].

Differences in findings are partly due to heterogeneity between studies. In particular what constitutes a portion of fish varies from study to study and has been shown to range from 85 g to 200 g depending on the type of fish as well as the country of the study [15,27,36]. In addition, categories of intake differ from study to study with some choosing very high or low categories of intake. We chose to assess intake from a more public health relevant context, but this resulted in very small numbers in the high consumption category (>2 portions/week), which limited the power to detect a true association. Furthermore, it is unclear whether the timing of exposure has any effect on outcomes and to our knowledge; no study to date has looked at all trimester specific fatty fish intakes in relation to birth outcomes. Of the studies which have assessed intake in more than one trimester and/or prior to pregnancy [16,20,23,26], one found a positive association with size at birth in overweight women for intakes before pregnancy but not in the final period of pregnancy [26]. Another found an increased risk of LBW babies in women reporting no fish consumption in the 3rd trimester, but not in trimester 1 [16]. Results from a Danish study showed a decreased risk of preterm birth with increasing total fish consumption in both trimester 1 and 2 [20]. Finally one study found a negative association with size at birth and fish intake reported in the 1st trimester but not in the 2nd trimester [23]. None of these studies however looked at types of fish consumed. Moreover, the choice of confounders tends to be inconsistent across studies and since not only in the present study, but also in other studies, high fish consumption has been shown to be strongly related to a higher education level and more healthy lifestyle habits, any positive associations between fish consumption and birth outcomes may be partly due to residual confounding by lifestyle-related characteristics if studies have failed to take these into account in their analysis. Additionally, discrepancies in findings between countries may be a reflection of differences in dietary patterns. This heterogeneity makes it hard to compare results.

### 4.3. Strengths

As a unique feature of this study we had two sources of dietary intake available which allowed us to derive a study specific estimation of a portion of fatty fish rather than using the SACN estimation of 140 g/portion [28]. This may have given a truer picture of actual intake of fatty fish within a cohort of British pregnant women. Fatty fish intake was averaged to weekly consumption and then divided into categories. This was done so as to better reflect the current UK guidelines on fatty fish consumption for pregnant women and women trying to conceive, and to make the results more applicable in a public health context.

We assessed maternal fish intake at three time points covering a wide window of exposure and taking into account variations across trimesters. Furthermore, only self-reported fatty fish intake was accounted for in the questionnaire. Therefore, the relationship with fatty fish could be assessed, as previous studies have combined type of fish such as lean fish, shellfish and molluscs in their overall analysis, biasing the true effect. Halldorsson et al. (2007) found a negative association with size at birth [22] and Ramon et al. (2009) found that consumption of larger fatty fish ≥ twice/week (such as swordfish) compared to <once/month was associated with a higher risk of SGA, however the P for trend across categories of intake was not significant [18]. Other studies have not specifically identified fatty fish within their analysis and therefore findings cannot be explicitly compared.

In our study information was available for a wide range of confounders. The objective measurement of salivary cotinine samples meant that smoking, a significant confounder in relation to maternal fish intake and birth outcomes, was assessed accurately with a biomarker.

### 4.4. Limitations

The questionnaires used in this study were originally designed to assess caffeine intake in pregnancy and not dietary fatty fish consumption. However, the questionnaire was validated with reference to caffeine intake in pregnant women [37]; and other food related questions were comparable to other methods used in the assessment of fish. Despite intakes being self-reported and thus presenting the issue of possible under-reporting, fatty fish consumption was assessed prospectively in trimesters 1 and 2, reducing the potential for differential measurement (recall) bias.

An explanation for non-statistically significant findings with fatty fish intake and birth outcomes could be due to the number of women included in the analysis (*n* = 1208) compared to other large well known cohorts [15,17,20,22,23] as well as the low consumption of fatty fish reported in our cohort. We had limited power to detect small associations due to the low numbers in the high consumption category, especially in trimester 3 (*n* = 409). The original study of caffeine and birth outcomes planned to follow up women several weeks after delivery to investigate whether their caffeine metabolism had returned to normal, using a caffeine challenge. This proposed data collection was expensive. To reduce costs without introducing selection bias, all cases (SGA and LBW infants) were recruited for postpartum follow-up, but only a sample of controls were taken to be the next two births that were not SGA or LBW. However, previous studies with smaller cohorts have detected associations in relation to fatty fish intake [18,24], although these women consumed high intakes of fish due to their Mediterranean diets.

In this analysis, fresh tuna was considered a fatty fish and was therefore included in the estimation of fatty fish intake and portion size. UK guidance has recently changed to exclude fresh tuna from being considered a type of fatty fish [38,39]. However, our analysis included fresh tuna as a fatty fish, to facilitate comparison with previous studies. With the low intake of fatty fish within this cohort and the relatively low proportion of consumers reporting fresh tuna intake (*n* = 25) this is unlikely to have influenced results. Canned tuna was excluded in the estimation of fatty fish intake as it is not considered a fatty fish in the UK. However, it is important to acknowledge that canned tuna still contains levels of EPA and DHA, and with tuna being the second most consumed type of fish in our cohort, it could still be a contributor to total EPA and DHA intake. Nevertheless, the few studies that have looked at fish subtypes such as canned tuna in relation to birth outcomes have been inconclusive making it hard to elucidate the potential impact excluding it could have on our findings. Mendez et al. (2010) found in their Spanish cohort of 657 pregnant women that maternal consumption of canned tuna (more than 1 portion per week) was associated with a significantly increased risk of SGA (adjusted OR: 2.49, 95% CI: 1.04 to 5.97). However, another Spanish study [18] similar in size found that mothers consuming more than or equal to 2 portions per week of canned tuna had a lower risk of having infants who were SGA (adjusted OR: 0.3, 95% CI: 0.1, 0.8) compared to mothers in the lower consumption categories.

Finally, a major weakness within this cohort was the lack of objective measurement of self-reported fish consumption. This could have been validated using a biomarker, such as erythrocytes concentrations of n-3 fatty acids, to indicate accurate fish intake during pregnancy, which has been addressed in previous studies [18,19,22,23,24].

## 5. Conclusions

Within this UK cohort of low risk pregnant women there was a low prevalence of fatty fish consumption and no evidence of an association between fatty fish intake prior to or throughout pregnancy with size at birth or preterm birth after adjusting for confounding.

Ideally, large cohort studies focusing on types of fish as well as timing of exposure, with a particular focus on the preconception period, are needed to help improve the understanding of the relationship between maternal fish intake during pregnancy and birth outcomes.

## Figures and Tables

**Table 1 nutrients-11-00643-t001:** Self-reported fatty fish intake across pregnancy.

	*n* (%)	Mean (g)	95% CI
Fish intake (g/week) (consumers only):			
4 weeks before pregnancy (*n* = 1116)	648 (58.1)	123.5	115.1, 131.9
First trimester (*n* = 1114)	652 (58.5)	106.4	98.9, 112.9
Second trimester (*n* = 812)	466 (57.4)	107.4	98.2, 116.6
Third trimester (*n* = 409)	177 (43.3)	136.5	118.8, 154.1
Categories of intake 4 weeks before pregnancy *			
No fish	468 (41.9)	0	0
≤2 portions/week	491 (44.0)	67.6	64.8, 70.5
>2 portions/week	157(14.1)	298.1	286.7, 309.6
Categories of intake trimester 1			
No fish	462 (41.47)	0	0
≤2 portions/week	524 (47.0)	64.3	61.0, 67.7
>2 portions/week	128 (11.5)	278.9	267.1, 290.7
Categories of intake trimester 2			
No fish	346 (42.6)	0	0
≤2 portions/week	396 (48.8)	71.3	66.6, 75.7
>2 portions/week	70 (8.6)	311.8	291.9, 331.7
Categories of intake trimester 3			
No fish	232 (56.7)	0	0
≤2 portions/week	131 (32.0)	75.4	70.4, 80.4
>2 portions/week	46 (11.3)	310.5	279.0, 341.9

* Categories based on the UK recommendations of no more than 2 portions of fatty fish/week [28]. One portion of fish is 101 g.

**Table 2 nutrients-11-00643-t002:** Characteristics of mothers by fatty fish intake in the 1st trimester of pregnancy (*n* = 1114).

	No Fatty Fish	≤2 Portions/Week	>2 Portions/Week	*p* *
	(*n* = 462)	(*n* = 524)	(*n* = 128)	
Age (years) mean (SD)	28.5 (5.6)	30.8 (4.4)	31.7 (4.6)	0.0001
Pre-pregnancy BMI (kg/m^2^) mean (SD)	25.1 (5.3)	24.4 (4.3)	23.9 (5.3)	0.01
Total energy intake (kcal) mean (SD)	2109.4 (595.6)	2111.5 (614.3)	2183.5 (670.8)	0.8
Caffeine intake (mg/day) mean (SD)	223.3 (225.4)	159.9 (151.3)	190.6 (177.6)	0.0001
Alcohol intake: % non-drinkers (*n*)	28.3 (127)	16.4 (84)	20.0 (24)	0.0001
Smoker at 12 weeks % (*n*) **	26.8 (117)	9.1 (461)	10.6 (13)	0.0001
IMD most deprived quartile % (*n*)	41.1 (182)	21.7 (109)	19.1 (24)	0.0001
University degree % (*n*)	24.5 (113)	50.4 (264)	56.3 (72)	<0.0001
European origin % (*n*)	94.8 (437)	93.5 (489)	94.5 (121)	0.7
Primigravida % (*n*)	45.3 (209)	51.1 (267)	40.9 (52)	0.06
Baby’s gender: % male (*n*)	52.6 (243)	49.1 (257)	43.8 (56)	0.2
Gestational hypertension % (*n*)	1.1 (5)	1.9 (10)	1.6 (2)	0.6
Gestational diabetes % (*n*)	0.2 (1)	0.4 (2)	0 (0)	0.7
Past history of miscarriage % (*n*)	22.4 (102)	22.9 (119)	27.8 (35)	0.4

* *p*-Value using one-way ANOVA and Kruskal-Wallis for normally and non-normally distributed continuous variables respectively, and χ^2^-test & Fisher’s exact test for categorical variables. Significant difference at *p* < 0.05. ** Smoking status based on salivary cotinine concentrations: non-smoker < 1 ng/mL, passive smoker 1–5 ng/mL, current smoker > 5 ng/mL. Where numbers do not add up it is due to a small proportion of missing data. SD, standard deviation; BMI, body mass index; IMD, Index of Multiple Deprivation.

**Table 3 nutrients-11-00643-t003:** The relationship between maternal fatty fish intake 4 weeks before pregnancy and size at birth & preterm delivery.

		**Unadjusted Change** **(95% CI)**	***p*** *****		**Adjusted Change **** **(95% CI)**	***p*** *****
**Birth weight (g)**	***n***			***n***		
No fatty fish	468	0	0.3	426	0	0.7
≤2 portions/week	491	45.8 (−23.3, 115.0)		459	−17.9 (−75.3, 39.5)	
>2 portions/week	157	71.6 (−27.1, 170.3)		144	−35.7 (−115.6, 44.1)	
		**Unadjusted OR** **(95% CI)**	***p*** *****		**Adjusted OR **** **(95% CI)**	***p*** *****
**SGA (<10th centile) *****	**cases/*n***			**cases/*n***		
No fatty fish	70/468	1	0.3	67/444	1	0.6
≤2 portions/week	57/491	0.7 (0.5, 1.1)		55/473	1.0 (0.6, 1.5)	
>2 portions/week	23/157	1.0 (0.6, 1.6)		23/150	1.3 (0.8, 2.3)	
**Low birth weight (≤2500 g)**						
No fatty fish	21/468	1	0.8	19/426	1	0.3
≤2 portions/week	20/491	0.9 (0.5, 1.7)		19/459	1.9 (0.7, 5.3)	
>2 portions/week	5/157	0.7 (0.3, 1.9)		5/144	3.1 (0.8, 12.7)	
**Preterm birth (<37 weeks gestation)**						
No fatty fish	24/468	1	0.2	23/426	1	0.4
≤2 portions/week	17/491	0.7 (0.4, 1.3)		17/459	0.8 (0.4, 1.6)	
>2 portions/week	3/157	0.4 (0.1, 1.2)		3/144	0.4 (0.1, 1.5)	

* *p* for trend for categories of fish intake ** Adjusted for maternal pre-pregnancy weight, height, age, parity, ethnicity, salivary cotinine levels, caffeine intake, alcohol intake, education, gestation and baby’s sex in multivariable linear regression for continuous outcome and multivariable logistic regression for categorical outcomes. *** Takes into account maternal pre-pregnancy weight, height, parity, ethnicity, gestation and baby’s sex.

**Table 4 nutrients-11-00643-t004:** The relationship between maternal fatty fish intake during pregnancy and size at birth & preterm delivery.

	Trimester 1						Trimester 2						Trimester 3					
		Unadjusted Change (95% CI)	*p* *		Adjusted Change ** (95% CI)	*p* *		Unadjusted Change (95% CI)	*p* *		Adjusted Change ** (95% CI)	*p* *		Unadjusted Change (95% CI)	*p* *		Adjusted Change ** (95% CI)	*p* *
**Birth weight (g)**	***n***			***n***			***n***			***n***			***n***			***n***		
No fatty fish	462	0	0.3	422	0	0.1	346	0	0.2	316	0	0.3	232	0	0.3	218	0	0.8
≤2 portions/week	524	30.4 (−37.9, 98.7)		488	−58.4 (−115.1, −1.7)		396	75.3 (−6.5, 157.1)		371	−47.3 (−113.0, 18.4)		131	109.6 (−25.4, 244.6)		126	−35.6 (−139.9, 68.7)	
>2 portions/week	128	87.7 (−19.2, 194.6)		118	−64.0 (−151.1, 23.1)		70	42.3 (−103.4, 188.1)		64	−71.4 (−185.8, 43.13)		46	52.6 (−146.8, 251.9)		43	−21.8 (−169.0, 125.4)	
		**Unadjusted OR** **(95% CI)**	***p*** *****		**Adjusted OR **** **(95% CI)**	***p*** *****		**Unadjusted OR** **(95% CI)**	***p*** *****		**Adjusted OR **** **(95% CI)**	***p*** *****		**Unadjusted OR** **(95% CI)**	***p*** *****		**Adjusted OR **** **(95% CI)**	***p*** *****
**SGA (<10th centile) *****	**cases/*n***			**cases/*n***			**cases/*n***			**cases/*n***			**cases/*n***			**cases/*n***		
No fatty fish	69/462	1	0.2	65/436	1	0.3	60/346	1	0.6	56/328	1	0.6	74/232	1	0.9	24/218	1	0.8
≤2 portions/week	69/524	0.9 (0.6, 1.2)		68/506	1.2 (0.8, 1.8)		58/396	0.8 (0.6, 1.2)		58/385	1.1 (0.4, 1.7)		40/131	0.9 (0.6, 1.5)		10/126	1.1 (0.7, 1.9)	
>2 portions/week	11/128	0.5 (0.3, 1.0)		11/123	0.7 (0.4, 1.5)		12/70	1.0 (0.5, 1.9)		12/67	1.5 (0.7, 3.0)		5/46	1.0 (0.5, 2.0)		5/43	1.2 (0.6, 2.5)	
**Low birth weight (≤2500 g)**	**cases/*n***			**cases/*n***			**cases/*n***			**cases/*n***			**cases/*n***			**cases/*n***		
No fatty fish	23/462	1	0.3	21/422	1	0.4	23/346	1	0.3	21/316	1	0.2	26/232	1	0.5	69/220	1	0.2
≤2 portions/week	21/524	0.8 (0.4, 1.5)		20/488	2.0 (0.7, 5.6)		17/396	0.6 (0.3, 1.2)		17/371	3.0 (0.9, 9.7)		10/131	0.7 (0.3, 1.4)		40/129	2.4 (0.6, 9.7)	
>2 portions/week	2/128	0.3 (0.1, 1.3)		2/118	1.2 (0.2, 7.4)		3/70	0.6 (0.2, 2.2)		3/64	1.5 (0.3, 8.1)		5/46	1.0 (0.4, 2.7)		15/45	5.5 (0.9, 31.9)	
**Preterm birth** **(<37 weeks gestation)**	**cases/*n***			**cases/*n***			**cases/*n***			**cases/*n***			**cases/*n***			**cases/*n***		
No fatty fish	26/462	1	0.05	25/422	1	0.2	21/346	1	0.06	21/316	1	0.2	18/232	1	0.3	18/218	1	0.6
≤2 portions/week	16/524	0.5 (0.3, 1.0)		16/488	0.6 (0.3, 1.3)		10/396	0.4 (0.2, 0.9)		10/371	0.5 (0.2, 1.2)		5/131	0.5 (0.2, 1.3)		5/126	0.6 (0.2, 1.7)	
>2 portions/week	2/128	0.3 (0.1, 1.1)		2/118	0.3 (0.1, 1.3)		4/70	0.9 (0.3, 2.8)		4/64	1.1 (0.4, 3.6)		3/46	0.8 (0.2, 2.9)		3/43	0.7 (0.2, 2.8)	

* *p* for trend for categories of maternal fish intake in linear and logistic regression models for continuous and dichotomous outcomes respectively. ** Adjusted for maternal pre-pregnancy weight, height, age, parity, ethnicity, salivary cotinine levels, caffeine intake, alcohol intake, education, gestation and baby’s sex in multivariable linear regression for continuous outcome and multivariable logistic regression for categorical outcomes. *** Takes into account maternal pre-pregnancy weight, height, parity, ethnicity, gestation and baby’s sex. LBW, low birth weight; *n*, number; OR, odds ratio; SGA, small for gestation age.

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
