# Peer review of "Maternal Fatty Fish Intake Prior to and during Pregnancy and Risks of Adverse Birth Outcomes: Findings from a British Cohort"

_nutrients, 2019, doi:10.3390/nu11030643_

Round 1

Reviewer 1 Report

This is a paper with an interesting concept of tracking fatty fish intake over different time points throughout preconception and pregnancy and relating it to pregnancy outcomes. Unfortunately, it appears that canned tuna was not included as a "fatty fish," when "servings of fatty fish" are at the core of the analyses. It appears as though in the data collection, women were told to only consider servings of tuna from fresh, not canned, sources in their estimate of their total fatty fish intake. This, in my mind, may be a fatal flaw in the study. I read the portion of the SACN report that said not to consider canned tuna a fatty fish because the processing lowers the fat content too much, but even so canned tuna is still a potent and very accessible source (at least in the US) of fatty fish and related nutrients, i.e. EPA+DHA. Can you provide more information about this decision and what the intake of tuna is in the UK? 

I also am wondering why the group taking EPA+DHA supplements were not analyzed as a comparator group to the fatty fish servings groups. Is this still possible? The other analysis I would be interested in seeing is the gestational age as a continuous variable as well as early preterm birth (<34 weeks gestation) as a categorical variable. 

Still, the fish intake question makes me question all the findings because it cuts out a potentially large source of EPA+DHA in women's diets, which is one of the most important nutrients that fatty fish provide and relates directly to pregnancy outcomes. If I'm incorrect about how the data were collected, please let me know and I would be happy to reconsider this paper after reanalysis. 

Reviewer 2 Report

Nutrients-448929 Maternal fatty fish intake prior to and during pregnancy and risk of adverse birth outcomes: findings from British cohort. 

This is a very interesting manuscript providing much needed data on the relationship between the intake of fatty fish during pregnancy and the risks of SGA, LBW and preterm birth. The study is well-performed and the manuscript is well-written. 

Major comments:

1.    Methods, Statistical analysis: Was maternal gestational weight gain measured in this study? It would have been useful to include this as a variable in the multivariate statistics given that weight gain in pregnancy may also be an important predictor for birth weight of the baby. 

2.    Methods, Statistical analysis: Did any of the mothers develop GDM or Preeclampsia? How was data from these women handled in the analysis? The methods section only describes those with previous high-risk pregnancies. Please also specify what the average gestational age at delivery was for the babies that were born preterm. 

3.    Results, estimation of portion size (24hr recall), l166-8: Please clarify how the fatty fish intake during the 1stor 2ndtrimester was calculated from the 24 hr recall data. Currently it states 13% in trimester 1 and 12% in trimester 2 with quite different numbers of women who did the recall. I am just not clear on how the number of 106 women was derived. 

4.    Results, frequency of fatty fish consumption, l179-181: What proportion of the women continuously consumed fatty fish and what proportion did not consume any fatty fish? Are these women that either did not consume any fish or the highest amount of fish throughout pregnancy different with regard to any of the biochemical/clinical/demographic characteristics? The data is currently only supplied for the first trimester of pregnancy and analysed per trimester but not for the pregnancy overall. 

5.    Results: is grams of fish intake correlated with income/socio-economic status at all? It appears that fish intake is just a measure of many factors that are determined by socio-economic status. This is supported by the fact that after adjusting for confounders, there is no relationship any more. 

6.    Results, table 2: the proportion smokers seems very high (between 70-80%): is this a true reflection of this cohort? 

7.    Results, Pregnancy outcomes, l235-237: to what extent is there an overlap between preterm babies and LBW babies? To what extent were the preterm babies spontaneous preterm deliveries versusthose delivered for indications such as maternal or fetal well-being? 

8.    Results, relationship between fish intake pre-pregnancy and birth outcomes: please rewrite lines 239-240 to make this sentence logical. At the moment, it is unclear if there is or is no evidence of an association between fatty fish intake and preterm delivery or birth size. From Table 3, this becomes clear but the sentence needs to be altered. 

9.    Results, table 3 and 4: it would be more informative to combine tables 3 and 4 into one table so the effects can be more easily compared. In that table, it would be good to include the total number of babies in each category of fish consumption that were born SGA, with LBW or preterm. 

Minor comments

1.    Discussion, lines 326-329: these seem to be in different size font/contain bolding? 

2.    Discussion, line 378: “of possible under-reporting” appears to be in bigger font. 

Round 2

Reviewer 1 Report

Thank you for your response. I understand that tuna is not considered a fatty fish in the UK based on the government’s definition. It’s concerning to me that this distinction appears to be based on analysis of one kind of canned tuna in either brine or oil in the UK nutrient database, which are not defined by their tuna species. Both entries are likely light chunk tuna (based on their reported n-3 PUFA levels) and are the kind with the lowest amount of fat and n-3 PUFA in both databases. There are 17 entries for different kinds of tuna and preparation methods in the USDA Nutrient Database. Per 100 g serving, light chunk tuna has 224 mg EPA+DHA, white or albacore canned tuna has 832 mg EPA+DHA, raw bluefin tuna has 1200 mg EPA+DHA, raw skipjack tuna has 256 mg EPA+DHA, and raw yellowfin tuna has 100 mg EPA+DHA. So, there is a 10-fold variation in EPA+DHA content in “tuna”! Yet the UK Nutrient Database just lists “canned tuna” and “fresh tuna” and then makes its distinction of “fatty fish” based on very few analyses and undefined species. Moreover, in 2017 tuna was tied with cod for the highest volume of fish sold in commercial foodservice in the UK (https://www.worldfishing.net/news101/regional-focus/uk-seafood-consumption-crisis), so this is even more disconcerting to me.

So, beyond this study, it’s concerning to me that the most commonly consumed fish in the UK that contains a decent level of EPA+DHA is not being counted in this (or any) dietary intake study of fatty fish in the UK. However, I digress…

Revisions Needed:

·      Can you clarify throughout which analyses (tables) were based on the frequency questionnaire data and which were based on the 24-hr dietary recall data (that was split into categories of fish intake)?

·      The abstract has apparently conflicting sentences about the primary outcomes of this study and needs clarification:

o   “Following exclusion of women taking cod liver oil and/or omega-3 supplements, fatty fish intake was related to size at birth and preterm delivery (<37 weeks gestation) in multivariable regression models adjusting for confounders including salivary cotinine.”

o   “After adjusting for confounders, no association was observed between fatty fish intake before or during pregnancy with size at birth and preterm delivery.”

·      Limitations (Lines 392-397)

o   “With the low intake of fatty fish within this cohort and the relatively low proportion of consumer reporting fresh tuna intake (n=25) this is unlikely to have influenced results.”

§  Due to the low number of women in the highest “fatty fish” intake group, I don’t believe that removing a type of fish from the fatty fish options that was the 2nd most consumed fatty fish in your recall data would not have a chance to influence results. Please acknowledge in your limitations.

Author Response

1.     Can you clarify throughout which analyses (tables) were based on the frequency questionnaire data and which were based on the 24-hr dietary recall data (that was split into categories of fish intake)?

§  Frequency of fish consumption derived from the questionnaires was converted to times per week, which was then multiplied by the portion estimate of fish obtained from the recall data in order to get weekly consumption in grams for each of the trimesters. (Please see manuscript lines 110-112)

2.     The abstract has apparently conflicting sentences about the primary outcomes of this study and needs clarification:

§  “Following exclusion of women taking cod liver oil and/or omega-3 supplements, fatty fish intake was related to size at birth and preterm delivery (<37 weeks gestation) in multivariable regression models adjusting for confounders including salivary cotinine.”

§  “After adjusting for confounders, no association was observed between fatty fish intake before or during pregnancy with size at birth and preterm delivery.”

§  Thank you we have slightly rephrased the text to clarify. Please see manuscript lines 26-27.

  Limitations (Lines 392-397)

3.     “With the low intake of fatty fish within this cohort and the relatively low proportion of consumer reporting fresh tuna intake (n=25) this is unlikely to have influenced results.”

§  Due to the low number of women in the highest “fatty fish” intake group, I don’t believe that removing a type of fish from the fatty fish options that was the 2nd most consumed fatty fish in your recall data would not have a chance to influence results. Please acknowledge in your limitations.

§  Thank you we have acknowledged this. Please see manuscript lines 397-401.